# Novel Rubber Composites Based on Copper Particles, Multi-Wall Carbon Nanotubes and Their Hybrid for Stretchable Devices

**DOI:** 10.3390/polym14183744

**Published:** 2022-09-07

**Authors:** Vineet Kumar, Siraj Azam, Md. Najib Alam, Won-Beom Hong, Sang-Shin Park

**Affiliations:** School of Mechanical Engineering, Yeungnam University, Gyeongsan 38541, Korea

**Keywords:** piezo-electric energy-harvesting device, stretchable devices, silicone rubber, multi-wall carbon nanotube, copper particles

## Abstract

New technologies are constantly addressed in the scientific community for updating novel stretchable devices, such as flexible electronics, electronic packaging, and piezo-electric energy-harvesting devices. The device promoted in the present work was found to generate promising ~6V and durability of >0.4 million cycles. This stretchable device was based on rubber composites. These rubber composites were developed by solution mixing of room temperature silicone rubber (RTV-SR) and nanofiller, such as multi-wall carbon nanotube (MWCNT) and micron-sized copper particles and their hybrid. The hybrid composite consists of 50:50 of both fillers. The mechanical stretchability and compressive modulus of the composites were studied in detail. For example, the compressive modulus was 1.82 MPa (virgin) and increased at 3 per hundred parts of rubber (phr) to 3.75 MPa (MWCNT), 2.2 MPa (copper particles) and 2.75 MPa (hybrid). Similarly, the stretching ability for the composites used in fabricating devices was 148% (virgin) and changes at 3 phr to 144% (MWCNT), 230% (copper particles) and 199% (hybrid). Hence, the hybrid composite was found suitable with optimum stiffness and robust stretching ability to be useful for stretching electronic devices explored in this work. These improved properties were tested for a real-time stretchable device, such as a piezoelectric energy-harvesting device and their improved voltage output and durability were reported. In the end, a series of experiments conducted were summarized and a discussion on the best candidate with higher properties useful for prospective applications was reported.

## 1. Introduction

The energy demands are increasing day by day and there is a sudden increase in this demand in the last few decades. With the limitation of resources, meeting these demands is becoming difficult to fulfill. Last century, most of the energy production was made through oil, petroleum and coal. However, these sources are limited and non-renewable and their use has created various environmental damages, such as global warming. These problems, such as global warming results in the melt of icebergs and the earth are slowly sinking into the oceans.

Thus, scientists around the globe are working on providing alternative routes of energy that are eco-friendly and mostly renewable. These green routes for producing and storing energy are super-capacitors [1], solar cells [2], batteries [3] and stretchable piezoelectric energy-harvesting devices [4]. Among them, batteries involve the use of acid that degrade the environment, while solar cells also have environmental issues.

However, capacitors and energy-harvesting devices are the cheapest, clean, and highly durable to produce or store energy [1,4]. Piezo-electric energy-harvesting devices or stretchable devices are promising sources of renewable energy sources and are thus explored in the present work. Piezo-electric materials include a family of materials ranging from crystalline materials, such as quartz-analogous crystal [5], ceramics materials, such as PZT [6], lead-free ceramics, such as barium titanate [7], semiconductors [8], and dielectric polymers [9].

Among them, PZT was found to be the most promising piezo-electric material [10] but due to its poisoning effect, its use is limited [11]. Thus, new lead-free materials were used especially based on dielectric polymers as practiced in this work [12]. These polymer-based stretchable devices are composed of a flexible substrate and electrode [13]. Both electrodes and substrate are made up of elastomer matrix namely silicone rubber, which is soft and has the high stretchable ability as desired. The electrode is generally made up of conductive composites, which are generally composed of carbon nanotubes mixed with silicone rubber [14] while the substrate is made up of virgin or filled composites [15].

The piezo-electric effect based on dielectric polymer and the mechanism of energy harvesting from the piezo-electric device is shown in Figure 1. The basic principle of piezo-electricity involves the development of electrical energy via mechanical deformation. When a mechanical strain is applied to piezo-electric material, it generates opposite charges on electrodes separated by the dielectric polymer substrate. The basic mechanism involves the generation of more and more opposite charges with more duration of the mechanical strain of the piezo-electric device and hence produces more voltage as described in Figure 1.

The polymer composites are made up of different types of polymers, such as elastomers [16] or thermosets [17]. The elastomers are frequently used for stretchable devices. These elastomers are made of natural rubber [18] or synthesized rubber, such as diene rubbers [19] or silicone rubber [20]. Among them, silicone rubber is most promising due to its ease of process, ease of cure, and hardness of around 40 [21].

These properties make silicone rubber a promising candidate for soft and stretchable devices, such as actuators [22] and strain sensors [23]. Virgin rubber has poor properties and is sticky and not useful for any applications. Therefore, curing is performed to improve the composites’ general properties (mechanical properties). Still, these mechanical properties are not enough to be suitable for practical applications. Thus, traditional fillers, such as carbon black were added while improving not only mechanical properties but also the thermal and electrical properties of rubber composites [24].

However, the carbon black used in these composites in high amounts (notably > 60 phr) and high loading alters the viscoelastic properties of rubber composites [25]. To fix this issue, nanofillers are employed in the last few decades that show robust improvement in properties at lower loading without altering the viscoelasticity of the composites. These nanofillers are clay minerals [26], silica [27], graphene [28] and carbon nanotube [29].

Among them, carbon nanotube was found utmost promise due to its favorable morphology and high aspect ratio [30]. Thus, the carbon nanotube is used in the present work as a nanofiller and high properties were reported at as small as 2 phr loading in the rubber matrix [31]. In addition to the carbon nanotube, copper particles are added to improve the overall properties of the composites [32]. Copper is considered a high-performance candidate for various applications due to its outstanding physio-mechanical properties. Due to different particle sizes of copper particles ranging from nanometer range to micron range. They can be used in rubber matrix as a reinforcing agent for improving various engineering applications.

Various studies demonstrate that copper is an outstanding material for improving electrical and thermal properties [33,34,35,36,37]. For example, Wang et al., report that copper can be used as a thermally conductive material in polybenzoxazine-based composites when added with boron nitride [33]. Their study further reports that copper can be successfully used to solve heat diffusion problems in electronic packaging [33]. Further studies by Yin et al., show that the copper nanowires introduced in boron nitride nanosheets can be useful for thermal management problems in flexible electronics [34].

Yang et al., studied the binary fillers, such as copper and tin in PVDF and improved the electrical conductivity by an order of magnitude as reported [35]. Moreover, Boudenne et al., studied the electrical and thermal behavior of the PP filled with two different types of copper particles and improved properties are reported [36]. They investigate the effect of particle size of copper particles and higher heat transport; an effective filler percolation threshold was attained from copper with small particle size [36].

In another study by Kumlutas et al., the effect of the shape of copper particles on thermal properties was noticed [37]. In their study, the copper particles with platelet or spherical or short fibers shapes were added to polyamide to study the orientation effect on the thermal properties of the composites [37]. The copper particles with shapes in form of fibers were found to be effective in enhancing thermal properties [37].

However, studies on copper particles in hybrid with MWCNT in silicone rubber for different types of mechanical properties, such as stretching ability, tribology, etc. are not yet reported and especially for piezoelectric energy harvesting applications. In our previous study, Mannikkavel et al., investigate the piezoelectric energy-harvesting device for MWCNT electrodes and HTV-RTV silicone rubber but only up to 1 V, and durability of 50,000 cycles was noticed [14].

One of the key limitations of our previous work was the poor voltage stability and durability of the device [14]. In this work, the substrate of the piezoelectric energy-harvesting device was filled with 1 phr of copper, MWCNT, or their hybrid, and their output voltage was monitored. The novelty in this work involves the use of copper, MWCNT or their hybrid as a reinforcing material in dielectric silicone rubber. The use of copper in single and hybrid forms is advantageous in piezoelectricity in the present work because it improves the electrical, mechanical, and thermal properties.

Mechanical properties, such as the stretching ability of piezo-electric devices are often ignored in studies but they are of great importance and studied in the present work. With the addition of MWCNT, the electrical properties are improved significantly but the stretching ability is greatly suppressed. It is because the addition of MWCNT promotes enhancing cross-linking density thereby making the composite stiff, and fragile and cracks are formed at an early stage of deformation thereby leading to the falling of voltage when used as electrodes in energy-harvesting devices [14].

Thus, we need a material that can improve stretching ability and maintain electrical conductivity with MWCNT to obtain robust performance in energy harvesting in stretching devices. For that, copper is ideal to achieve high performance by enhancing stretching ability and optimum crosslinking density without harming electrical properties as obtained by MWCNT in the electrode. From experiments, the output voltage was as high as ~6 V, and durability for MWCNT-based substrate was >0.4 million cycles. Thus, the present work was advantageous in terms of the amount of output voltage, stability of voltage and high durability.

## 2. Materials and Methods

### 2.1. Materials

The RTV silicone rubber with the commercial name “KE-441-KT” with a transparent appearance was obtained from Shin-Etsu Chemical Corp. Ltd. (Tokyo, Japan). It was used as an elastomeric matrix for the present work. The vulcanizing agent “CAT-RM” was obtained from Shin-Etsu Chemical Corp. Ltd. The multi-wall carbon nanotube (MWCNT) was used as a reinforcing nanofiller. The MWCNT has a diameter of <15 nm, lateral dimensions of 500 nm–1 μm and thus a high aspect ratio of >65. The chemical purity of the MWCNT was >95% and with a commercial name of “CM-100” and purchased by Hanwha Nanotech Corporation Ltd. (Seoul, Korea). The micron-size copper particles were obtained from Duskan Reagents (Ansan-si, Korea). The mold-releasing agent was obtained from Nabakem (Pyeongtaek-si, Korea).

### 2.2. Preparation of Composites

The fabrication of novel rubber composites was started following the previous work instructions [14]. The optimized route involves the preparation method of spraying the molds with a mold-releasing agent and keeping them drying at room temperature for 2–3 h. Then, the liquid state of RTV-SR (without the use of solvent) was poured into a beaker and a known amount of nanofillers was added (Table 1) to RTV-SR and mixed thoroughly for up to 10 min.

After the filler-rubber mixing phase, the 2 phr of the vulcanizing agent were added to the composite and mixed for 1 min before pouring them into the sprayed molds. These molds were cylindrical (10 × 20 mm) for compressive properties or rectangular (2 × 60 × 60) for tensile properties. The molds were then manually pressed and kept at room temperature for curing for 24 h. Then, the samples were taken out of molds and tested for improved properties and novel applications, such as piezoelectric energy harvesting.

### 2.3. Characterization Technique

The morphological features and dispersion of filler were together studied through SEM microscopy (S-4800, Hitachi, Tokyo, Japan). The composite samples were sliced into 0.5 mm thick samples through a surgical blade and mounted onto an SEM stub before coating. Both powder samples and composite samples were sputtered with platinum coating for at least 2 min to make the surface electrically conductive for SEM examinations. The crosslink densities of rubber composites were calculated based on Flory–Rehner equation [38] as
(1)Vc=−[ln(1−Vr)+Vr+χVr2]Vsdr(Vr13−Vr2)
where V_c_ denotes the crosslink density, χ = 0.465 is the interaction parameter for silicone rubber and toluene system [39], V_s_ = 106.2 is the molar volume of swelled solvent toluene, d_r_ is the density of the rubber, and V_r_ is the volume fraction of rubber in the swollen state. The volume fraction of rubber was calculated from equilibrium swelling data for 7 days in toluene. The V_r_ was calculated as—
(2)$$V_{r}=\frac{\left(w_{r}/d_{r}\right)}{\left(w_{r}/d_{r}+w_{s}/d_{s}\right)}$$
where w_r_ and w_s_ are the weight of rubber and solvent, respectively; and d_r_ and d_s_ are the densities of rubber and solvent, respectively.

The compressive and tensile mechanical properties were examined through a universal testing machine (UTS, Lloyd Instruments, Bognor Regis, UK). The compressive mechanical properties were tested on cylindrical samples (20 × 10 mm) at a strain rate of 2 mm/min and up to 35% maximum strain. This strain value was selected because the sample fractured at a higher strain. The tensile mechanical properties were tested on dumbbell-shaped samples with a gauge length of 25 mm and thickness of 2 mm. The strain rate of tensile tests was maintained at 200 mm/min.

These mechanical measurements were obtained according to DIN 53 504 standards. The tribometer used in the present work was obtained from HM Hanmi Electronics, Korea. The model is “STM Smart”, and the model name is “universal material testing machine”. The experiments were performed at a load of 5 N, frequency of 3 Hz and distance of up to 50 m. The experimental set up and dimensions of pin used in experiments were described in Figure 2. The sample dimensions are 25 cm × 10 cm, and the thickness is 0.8 cm.

The output voltage and durability of the stretchable device were performed by a mechanical testing machine (Samick-THK, Daegu, Korea) under cyclic loading. The optical image, dimension of the sample, area of the electrode, dimension of substrate and electrode, and other useful information were described in the previous study [14]. The measurements were performed on an 8 mm thick substrate sandwiched by a 0.2 mm thick electrode in the stretchable device. The substrate was made up of different composites filled with 1 phr of copper, MWCNT or their hybrid and the electrode was made up of 2 phr MWCNT and 2 phr MoS_2_. MoS_2_ was added to improve the fracture strain of the MWCNT as optimized in a previous study [21].

## 3. Results and Discussions

### 3.1. Morphology of Filler Particles

The morphology of the filler particles in composites significantly influences the properties [40]. The filler with favorable morphology is dispersed easily and uniformly and leads to improved properties. Thus, it is important to study the morphology of the filler in the present study through SEM into a number of samples and their representative images are reported. Figure 1a shows the morphology examination of MWCNT. MWCNT has 1-dimensional (1-D) morphology with a tube shape appearance and is a nanofiller since its dimensions are in nm scale.

Since its invention by S. Iijima in 1991 [41], it has become a promising candidate for filler, and various studies prove that it is a promising reinforcing filler and a candidate for improving electrical and thermal properties due to its high aspect ratio [42]. Moreover, its high surface area allows higher stress transfer from polymer chains to filler particles due to higher interfacial area and their interactions [43]. The copper particles are of irregular shape and 3-D in nature as shown in Figure 1b. The copper particles are micron-sized particles and usually have poor reinforcing properties as compared with MWCNT. Its small surface area is also responsible for its poor mechanical, electrical and thermal properties when used as filler in rubber composites. The dispersion of filler also influences the properties significantly and studied in Figure 2.

### 3.2. Filler Dispersion of Composites through SEM Microscope 

In this work, filler dispersion is studied through an SEM microscope. Multiple images per sample were recorded and their representative images at different resolutions are presented in Figure 2. All the samples show uniform dispersion of filler with no signs of aggregations. However, the virgin samples show a neat rubber matrix with no presence of filler as excepted. The samples filled with MWCNT show that the MWCNT particles form long-range filler networks in the composite.

This is attributed to the higher aspect ratio of MWCNT (65) that allows continuous network formation in the composite. These features support the higher properties of the MWCNT-based composites [42,43]. Moreover, the interfacial interactions between MWCNT particles and polymer chains in rubber matrix are good, which also support higher properties of MWCNT composites.

The higher filler–polymer interfacial interactions are supported due to the higher interfacial area, which is provided by the higher surface area of MWCNT. In copper-based composites, since the particle size is in the micron range and the copper has a poor aspect ratio, the dispersion of copper is scarce in the rubber matrix due to the presence of a smaller number of copper particles in the rubber matrix. In some cases, the polymer chains are adsorbed on the copper particles, which expect higher reinforcement but due to the scarcity of several copper particles, the reinforcement excepted is low as compared to MWCNT composites.

However, in the case of hybrid composites, the SEM micrographs show that the polymer chains are not only adsorbed on micro copper particles but the MWCNT particles are also found in the vicinity of these copper particles. This behavior could result in synergism among the binary filler particles with improved reinforcing properties. In a few cases, the hollow structures are formed near copper particles while in some cases the polymer chains were found adsorbed on copper particles along with the MWCNT particles and this behavior affects the properties of the hybrid composites.

The features, such as the formation of the hollow structures around copper particles in the rubber matrix, mimic the sinking of solid copper particles in polymer solution with poor interfacial activity. These features support poor reinforcing effects, especially when compared with MWCNT as only fillers in composites. The XRD data of the hybrid composite to show that copper particles and MWCNTs are properly embedded in the rubber matrix is presented in Appendix A.

### 3.3. Cross-Linking Density of Filled Composites

The cross-linking density of different composites at 3 phr is presented in Figure 3. We found from the experimental testing that the crosslink density was higher for hybrid composites and highest for MWCNT-based composites while lowest in copper composites. The higher cross-linking density in MWCNT is due to its nanoscale dimensions, which help the even dispersion of the curatives in the rubber matrix.

Another feature of MWCNT is its higher surface area, which also allows the better dispersion of curatives in rubber matrix [44]. The poor cross-linking density of copper is due to its micron range size and poor surface area, which leads to uneven dispersion of curatives and leads to a poor density of curatives. These measurements were correlated with the mechanical properties of composites in subsequent sections of the manuscript.

### 3.4. Mechanical Properties

The mechanical properties, such as mechanical stretching ability or stretch until fracture, of composites, are vital to be studied for stretchable devices [45] studied in this work. The mechanical stretchability of a device depends upon several parameters, such as filler dispersion [46], the morphology of filler [47], an aspect ratio of the filler [48], the type of rubber matrix used [49], or the type of mechanical deformation [14]. Here, compressive and tensile strain were applied, and the mechanical behavior of the stretchable device under different strains were explored.

#### 3.4.1. Under Static Compressive Strain

The mechanical behavior of the composites under compressive strain was studied and presented in Figure 4a–c. We found that the compressive stress increases with an increase in compressive strain. It is attributed to the increase in packing density [50] of polymer chains and filler particles, which increases with an increase in compressive strain. Moreover, it was interesting to note that the compressive stress increases linearly up to 15% compressive strain and then increases exponentially up to 35%, especially in the case of MWNT composites in Figure 4a.

Such an increase is due to the higher anisotropic behavior of MWCNT and its higher aspect ratio [48] than copper particles. In addition to this, it was also interesting to note that with an increase in filler content from 1 to 3 phr, the compressive stress increases for all fillers. It is attributed to the formation of filler networks [51] and enhanced filler–polymer and filler–filler interactions [52] that lead to induced stiffness making the composite tough and highly reinforced.

In Figure 4d, the behavior of compressive modulus as a function of filler content is proposed. We found that the compressive modulus increases with an increase in filler content. It is due to improved interfacial interaction among filler and rubber matrix [52,53], which increases with the increase in filler volume fraction in the composite. In addition, it is noteworthy that up to 1 phr, the MWCNT, and hybrid composites show the highest compressive modulus while after 1 phr, MWCNT leads to higher values till 3 phr. It is attributed to higher filler anisotropy and aspect ratio of MWCNT, which make it a promising reinforcing agent for rubber matrix [48].

The higher modulus for hybrid composites at 1 phr is due to synergism among MWCNT and copper particles. After 1 phr, the modulus was lower than MWCNT while higher than copper particles. The lowest modulus of copper particles is due to its poor aspect ratio of nearly 1, poor reinforcing effect, and poor interfacial interaction due to small surface area and particle size in the micron range. The poor mechanical properties of copper are also due to a decrease in crosslinking density of the curatives with the addition of copper particles as justified in Figure 3.

#### 3.4.2. Under Static Tensile Strain

The behavior of stress–strain profiles under tensile strain was reported in Figure 5a–c. Particularly, these measurements help us to determine properties, such as the modulus, tensile strength, and fracture strain of composites [54], as presented in Figure 5d–f. We found from stress–strain curves that the tensile stress increases with increasing tensile strain until fracture strain. Such a property can be stated as improved interactions in composites, such as cross-linking density [55] and other interfacial interactions [52,53].

With an addition of filler in the rubber matrix, the tensile stress increases with increasing filler content in the composite. This can be attributed to improved filler–filler and polymer-filler interactions in the composites [53]. It is also interesting to note that the mechanical properties except fracture strain were highest for MWCNT-based composites. It is attributed to the higher anisotropy of MWCNT that allows it to form percolative networks at lower filler loading of approx. 2 phr and higher mechanical properties, such as tensile modulus and tensile strength was reported.

The behavior of tensile modulus (Figure 5d), tensile strength (Figure 5e), and fracture strain (Figure 5f) of different composites [54] were studied and reported. We found that the tensile modulus increases with an increase in filler loading for MWCNT-based composites and was lower for hybrid and copper particles. Such an increase in modulus for MWCNT is attributed to the higher anisotropic effect of MWCNT particles [56] with improved interfacial interaction [52,53] and higher crosslink density resulting in a higher modulus.

However, the tensile modulus is lower for hybrid and copper-based particles, which is due to the poor reinforcing effect of copper particles due to the low anisotropic effect of filler, poor interfacial interaction, and lower crosslinking density. Similarly, the tensile strength was higher for hybrid composites up to 2 phr and higher for MWCNT at 3 phr. The higher tensile strength for hybrid composites is due to the synergistic effect and the higher tensile strength at 3 phr for MWCNT was due to the high anisotropic effect of MWCNT as a filler in composites [48].

In the end, the fracture strain was higher for copper and hybrid composites and lower for MWCNT-based composites. The higher fracture strain for a hybrid at 1 phr is due to the synergistic effect among the binary filler and the higher for copper composites is due to lower crosslinking density for copper-filled composites. The lower crosslinking density simulates the freer motion of polymer chains leading to higher fracture strain.

### 3.5. Theoretical Prediction of Hybrid Rubber Composites via Different Models

The theoretical prediction of modulus for one and two-component systems are well known in the literature [56]. These models generally depend on the aspect ratio of the filler, the volume fraction of the filler, and their interactive factors for a two-component system. Existing models, such as the Guth–Gold Smallwood model [56,57] and Halpin-Tsai theoretical equations [56,58] were used to study its deviation with the experimental results in the present study. After plotting the equations, it was found that the existing models fall within the experimental results, especially at lower volume fractions of the filler and the results are presented in Figure 6a,b.

The Guth–Gold Smallwood equation for two components system [56] can be written as—
E_1+2_ = E_o_ [(1 + 0.67f_1_ϕ_1_) + (1 + 0.67 f_2_ϕ_2_)]×i’(3)

Here, the E_1+2_ is the combined predicted modulus of the hybrid system. Then, ϕ_1_ and ϕ_2_ are the volume fraction of the filler of two components. “i” is the interactive interaction parameters of the model. f_1_ and f_2_ are the aspect ratio of the fillers. The above model can be produced by setting the interactive species in a range of “0.4–0.5”, which directly depends mainly on the volume fraction of the filler in the rubber matrix. The physical significance of the interactive factor is that the higher the value of the volume fraction of the filler particles and the higher the interaction among the filler particles and their interfacial interaction species. Similarly, the Halpin-Tsai theoretical equations for two components [56] can be written as—
E_1+2_ = E_o_ [(1 + 2 f_1_ϕ_1_)/(1 − ϕ1) + (1 + 2 f_2_ϕ_2_)/(1 − ϕ_2_ )]×i’(4)

Here, all the components are the same as the above models except “E_o_”, which means modulus of unfilled rubber. Both the models agree well with the experimental behavior of the composites up to 2 phr and then deviate. However, the Guth–Gold Smallwood model is closest to the experimental data. The interactive factor “i” for two components composite system was calculated and their corresponding values of modulus are reported in Figure 6c,d.

We found from the plots that the corresponding modulus increases with increasing interacting factors. It is also interesting to note that the interactive factor shifts as filler loading increases in the rubber matrix (Figure 6d). It is attributed to the synergistic effect of the two filler components in the composite. Moreover, the increases are due to filler–filler and interfacial interactions with increasing interactive factors in the composites. Here, the values of the interacting factor are in the range of 0.1 (poor interaction) to 1 (perfect interaction). However, experimentally, it is difficult to reach an interactive factor value of 1.

### 3.6. Tribology Properties of Rubber Composites

The tribological properties, such as the coefficient of friction of the rubber composites were studied and explored in the present study (Figure 7a,b). From the results, the properties were found to be correlated with the crosslink density (Figure 3) of the composites [44]. The copper with lower crosslink density was found to exhibit a higher coefficient of friction, which is higher than all composites including virgin samples. The experiments show that the best tribology properties, such as the lowest coefficient of friction were found in composites filled with MWCNT. The hybrid composite shows a coefficient of friction higher than MWCNT and lower than copper-based composites. Thus, a hybrid composite can be a candidate for optimum and medium tribological properties. The superior tribological properties of MWCNT are attributed due to its higher crosslinking density, high aspect ratio and higher polymer-filler interfacial interaction in the composite.

## 4. Industrial Applications

### 4.1. Voltage Output of Different Types of Stretchable Devices

The specimen thickness is 8 mm in the compression loading, and the load is applied for 4 mm at 2 Hz. All three specimens are of the same dimensions. Electrode thickness is 0.2 mm painted on both sides of the substrate. Copper tape is attached to the electrode to connect it to the digital multimeter to record the voltage output, which is obtained from the flexible specimen. Load is applied for one hour, which applies 7200 cycles of loading to the specimen. The constant amplitude of loading is maintained during the testing. During the testing, the electrode material is kept constant. The only change is done in the substrate materials. CNT + MoS_2_ binary filler reinforcement is chosen for preparing the electrode.

In the substrate, three different types of filler are reinforced in each specimen category. The performance of the substrate on reinforcing fillers with the polymer is studied in this experiment. When CNT is reinforced into the substrate, the voltage is around 1.25 V from the initial cycles. Due to the piezoelectric nature of the silicone polymer, the voltage is produced from the deformation. The geometry of the specimen is similar to the capacitor. Applying further loading during voltage production causes the activation of charge carriers from the substrate. The voltage production is kept constant until the end of 7200 repeated cycles for one hour of loading at 2 Hz. It represents the constant voltage production due to the activation of charge carriers in the steady state, which is represented in Figure 8a.

Figure 8b consists of a copper-reinforced substrate. Its voltage value in the initial loading is around 1 V. This value is suddenly decreasing due to the breakage of the electrode on initial repeated loading. Then, saturation occurs in the breakage, and a constant voltage is produced until the end of 7200 cycles. In comparison to the CNT-based substrate, the copper-based specimen can be able to produce less amount of voltage. Similarly, the hybrid specimen also produces a constant amount of voltage on repeated loading (Figure 8c). The voltage production from the hybrid specimen is slightly above 1 V. When compared with the CNT, CNT can produce more dense voltage than other materials. It is very clear from the obtained graphs of three different materials.

### 4.2. Durability of the Stretchable Device for the Best Candidate MWCNT Substrate

The specimen’s substrate thickness is 8 mm over 0.2 mm electrodes painted on both sides. MWCNT is added to the substrate. The electrode is made of reinforcing MWCNT and MoS_2_ nanoparticles. 21 mm hemispherical loader is used for applying load to the specimen. The flexible electrodes are connected to the multimeter to determine the produced voltage via the copper electrode. Over the 8 mm substrate, the deformation is applied for 50%, which is 4 mm. The load is applied at the frequency of 2 Hz.

On applying a deformation to the specimen, the piezoelectric specimen can be able to produce voltage output. The specimen can produce continuous voltage output during repeated loading cycles. During the initial cycles of loading, as increasing the number of cycles, the voltage is also getting increased. This is caused by charge carriers activating at an increasing rate [59]. After 8000 cycles, the voltage value is maintained constant, improving the number of cycles for up to 71,000. Then, the voltage value is slightly increasing due to the activation happening at the enhanced rate. The voltage value is slightly increasing for up to 135,000 cycles.

Again, the rise in voltage due to the activation occurs in the enhancing rate. After that, there is a sudden increase in voltage. It is mainly because of the electrodes’ charging for a long time. Applying load further causes voltage enhancement in a more significant amount [59]. Then the voltage is gradually increasing. The voltage value around 100 thousand cycles is less than 2 V. During 200 thousand cycles, the voltage value is around 3 V. 50% voltage value is improved from 100 thousand to 200 thousand cycles.

Again, the voltage value improves as the number of cycles increases. The maximum voltage obtained around 300 thousand cycles is less than 5.5 V. Increasing the number of cycles further causes the improvement of voltage, which reaches the maximum voltage value of around less than 6 V at 350 thousand cycles. After that, the improvement of loading causes the decrement of voltage. This is due to the breakage of the electrode due to repeated cycles. The breakage in the electrode causes the conductivity path breakdown in the electrode. It causes a reduction in the voltage generated from the specimen. The electrode cannot transport the voltage generated from the specimen due to its reduced conductivity. The voltage value was reduced up to 380 thousand cycles. This is due to the gradual breakdown of the electrode, causing the regular voltage reduction. Then saturation occurs in the electrode breakdown causing the stabilization of voltage production on repeated cycles. The voltage variation for 0 to 400 thousand cycles is shown in Figure 9a,d.

## 5. Conclusions

The present work develops and studies impressive voltage generation of around ~6 V in stretchable devices based on rubber composites with high durability of >0.4 million cycles. These rubber composites were obtained by solution mixing of RTV-SR and different types of nanofiller, such as MWCNT or copper particles with micron size and their hybrids. The dispersion of these particles was obtained from SEM micrographs and uniform dispersion was noticed. MWCNT shows 1-D tube-shape morphology with an aspect ratio of around 65, while copper particles were 3-D, the irregular shape and a low aspect ratio of around 1.

The mechanical stretchability and compressive modulus were studied and correlated with the stretchable device performance. For example, the compressive modulus was 1.82 MPa (virgin) and increased at 3 per hundred parts of rubber (phr) to 3.75 MPa (MWCNT), 2.2 MPa (copper particles), and 2.75 MPa (hybrid). Similarly, the stretching ability for the composites used in fabricating devices was 148% (virgin) and changes at 3 phr to 144% (MWCNT), 230% (copper particles), and 199% (hybrid).

In the end, the results were summarized and concluded and the properties were correlated with the high performance of the stretchable devices. In conclusion, this work addresses the methods to obtain high-performance stretchable devices with novel properties and applications, such as flexible electronics or electronic packaging. The work also supports the use of RTV-SR suitability to obtain desired flexibility and stretchability to be useful for engineering applications. This work also recommends that hybrid fillers can also be useful to obtain optimum stiffness and stretchability.

## Data Availability

Not applicable.

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
