# Peer review of "Novel Rubber Composites Based on Copper Particles, Multi-Wall Carbon Nanotubes and Their Hybrid for Stretchable Devices"

_polymers, 2022, doi:10.3390/polym14183744_

Round 1

Reviewer 1 Report

In the manuscript of polymers-1875812, the authors introduced a rubber composites based on copper particles and multi-wall carbon nanotube for stretchable devices, which was found suitable with optimum stiffness and robust stretching ability to be useful for stretching piezoelectric energy harvesting device. However, the research is only supplementary and lack of novelty, so I suggest it to be rejected. The following comments are provided for improvement.
1.  In the section of introduction, the Innovation needs further clarification.
2. The role of copper particles is fuzzy in the improvement of piezoelectric energy harvesting, so the working mechanism needs further study.

Author Response

Responses to the Editors and Reviewers’ comments for the Manuscript (polymers-1875812-R1)

We have now revised our manuscript based on the Editor and Reviewers’ valuable comments and suggestions. All changes and corrections are highlighted in RED color in the manuscript. Here, we hope that the Editor and Reviewers will be satisfied with our responses and thereby our revised manuscript. We are much thankful to the Editors and Reviewers for giving us a chance to improve the manuscript for publication in Polymers Journal.

Reviewer #1
In the manuscript of polymers-1875812, the authors introduced a rubber composites based on copper particles and multi-wall carbon nanotube for stretchable devices, which was found suitable with optimum stiffness and robust stretching ability to be useful for stretching piezoelectric energy harvesting device. However, the research is only supplementary and lack of novelty, so I suggest it to be rejected. The following comments are provided for improvement

Response: Thank you for your comment. However, the paper is improved as desired.

1) In the section of introduction, the Innovation needs further clarification.

Response: Thank you for the suggestion. The details on types of piezo-electric materials, mechanism of piezo-electricity, and novelty of the work are detailed as expected kindly by the reviewer. Please refer introduction part of the revised paper for changes.  

2) The role of copper particles is fuzzy in the improvement of piezoelectric energy harvesting, so the working mechanism needs further study.

Response: Thank you for the kind suggestion. Copper has a great role in affecting the properties of piezo-electric materials. The detailed mechanism and role of copper in piezoelectricity are provided in the introduction section. Please refer introduction part of the revised paper for changes.     

We again like to thank the reviewer for their critical suggestions and for improving the manuscript.

Reviewer 2 Report

1.      In the introduction, the basic principle of piezoelectric devices should be elaborated for broad readers of the Polymers journal. Some of the recent literature maybe helpful such as https://doi.org/10.1007/s10854-020-04148-2.

2.      It is true that copper can improve electrical and thermal properties, but how copper can play a role in piezoelectricity. The authors need to properly explain the role of copper particles in piezoelectricity production with proper references.

3.      XRD data of the hybrid composite is required to show that copper particles and MWCNTs are properly imbedded in the rubber matrix.

4.      Use proper equation numbers for all equations such as equations on page 4.

5.      The X-Axis title (Mixes) of Figure 3 is confusing. The authors need to revise Figure 3.

Reviewer 3 Report

The authors fabricated the rubber composites based on copper particles, multi-wall carbon nanotube, and their hybrid. They showed that the proposed composites are suitable with optimum stiffness and robust stretching ability with applications in stretchable electronics. The results are interesting and are well presented. This reviewer has one major concern: The methodology of fabricating rubber composite (based on copper particles or carbon nanotubes) is not new at all. It is hard to justify the novelty or major contribution of the work.

Author Response

Responses to the Editors and Reviewers’ comments for the Manuscript (polymers-1875812-R1)

We have now revised our manuscript based on the Editor and Reviewers’ valuable comments and suggestions. All changes and corrections are highlighted in RED color in the manuscript. Here, we hope that the Editor and Reviewers will be satisfied with our responses and thereby our revised manuscript. We are much thankful to the Editors and Reviewers for giving us a chance to improve the manuscript for publication in Polymers Journal.

Reviewer #3

The authors fabricated the rubber composites based on copper particles, multi-wall carbon nanotube, and their hybrid. They showed that the proposed composites are suitable with optimum stiffness and robust stretching ability with applications in stretchable electronics. The results are interesting and are well presented. This reviewer has one major concern: The methodology of fabricating rubber composite (based on copper particles or carbon nanotubes) is not new at all. It is hard to justify the novelty or major contribution of the work.

Response: Thank you for the comment and suggestion. Copper and MWCNT have a great role in affecting the properties of piezo-electric materials. The detailed mechanism and role of copper in piezoelectricity are provided in the introduction section. Please refer introduction part of the revised paper for changes.   

We again like to thank the reviewer for their critical suggestions and for improving the manuscript. 

Round 2

Reviewer 1 Report

Since the authors have addressed the reviewers' comments, the paper can be  recommended to be published in Polymers. 

Reviewer 3 Report

The paper can be published as it is since the authors have addressed the reviewer's comment satisfactorily.